# Cas13b-dependent and Cas13b-independent RNA knockdown of viral sequences in mosquito cells following guide RNA expression

Priscilla Ying Lei Tng[1,2], Leonela Carabajal Paladino[1], Sebald Alexander Nkosana Verkuijl[1,3], Jessica Purcell[1,4], Andres Merits[5], Philip Thomas Leftwich ![ORCID] [1,6], Rennos Fragkoudis[7,8], Rob Noad[2] & Luke Alphey ![ORCID] [1✉]

*Aedes aegypti* and *Aedes albopictus* mosquitoes are vectors of the RNA viruses chikungunya (CHIKV) and dengue that currently have no specific therapeutic treatments. The development of new methods to generate virus-refractory mosquitoes would be beneficial. Cas13b is an enzyme that uses RNA guides to target and cleave RNA molecules and has been reported to suppress RNA viruses in mammalian and plant cells. We investigated the potential use of the *Prevotella sp. P5-125* Cas13b system to provide viral refractoriness in mosquito cells, using a virus-derived reporter and a CHIKV split replication system. Cas13b in combination with suitable guide RNAs could induce strong suppression of virus-derived reporter RNAs in insect cells. Surprisingly, the RNA guides alone (without Cas13b) also gave substantial suppression. Our study provides support for the potential use of Cas13b in mosquitoes, but also caution in interpreting CRISPR/Cas data as we show that guide RNAs can have Cas-independent effects.

[1] Arthropod Genetics, The Pirbright Institute, Ash Road, Pirbright GU24 0NF, UK. [2] Pathobiology and Population Sciences, The Royal Veterinary College, Hawkshead Lane, Hertfordshire AL9 7TA, UK. [3] Department of Zoology, University of Oxford, 11a Mansfield Road, Oxford OX1 3SZ, UK. [4] Vector Biology Department, Liverpool School of Tropical Medicine, Liverpool L3 5QA, UK. [5] Institute of Technology, University of Tartu, Nooruse 1, Tartu 50411, Estonia. [6] School of Biological Sciences, University of East Anglia, Norwich Research Park, Norwich NR4 7TJ, UK. [7] Arbovirus Pathogenesis, The Pirbright Institute, Ash Road, Pirbright GU24 0NF, UK. [8] The University of Nottingham, School of Veterinary Medicine and Science, Sutton Bonington, Loughborough LE12 5RD, UK. ✉email: luke.alphey@pirbright.ac.uk

*A*edes aegypti and *Ae. albopictus* are vectors of multiple high impact diseases caused by viruses with positive-strand RNA genomes such as dengue, chikungunya (CHIKV) and Zika, which are major health burdens in many tropical and sub-tropical regions and for which there are no effective licensed vaccines[1–5]. CHIKV causes severe arthralgia, which is a burden on healthcare systems, and large-scale urban epidemics have been recorded as recently as 2019[6–10]. The development of new methods to reduce the capacity of this mosquito to transmit arboviruses would, therefore, be very valuable.

Prokaryotic clustered regularly interspaced short palindromic repeats (CRISPRs) and CRISPR-associated (Cas) proteins constitute a bacterial adaptive immune system against foreign nucleic acids such as those of bacteriophages. Upon infection, foreign nucleic acids are processed, and short fragments are integrated into the CRISPR array which is subsequently transcribed to produce guide RNAs. Different Cas proteins then use CRISPR-RNA guides to recognise specific DNA or RNA targets and cleave the complementary sequences[11]. Cas13 enzymes are RNA-guided ribonucleases from the class 2 CRISPR-Cas system subtype VI-B from Gram-negative bacteria[12–14]. The Cas13b enzymes from a range of different bacteria have been shown to be able to process their guides from longer RNAs and then use them to target and cleave RNA[12,13].

The Cas13b system has been proposed as an antiviral approach to target viral RNA in plants[15–17] and mammals[18]. Hence, we wanted to explore if Cas13b could be a useful tool to target viral RNA in mosquitoes.

Here we report the effect of the well-characterised Cas13b from *Prevotella sp. P5-125*[12] when used to target RNAs containing sequences from CHIKV in *Ae. aegypti* and *Ae. albopictus* mosquito cells. Two RNA guides were designed against the non-structural protein 2 (nsP2) region of CHIKV and tested in four different mosquito cell lines against (i) a chimeric firefly luciferase reporter plasmid containing the CHIKV sequence corresponding to nsP2 and (ii) a CHIKV split replication system[19,20] (Fig. 1). We demonstrate that Cas13b is capable of identifying viral RNA sequences in mosquito cells and potentially mediating the specific degradation of such sequences. Surprisingly, the guide RNAs are also able to induce degradation in the absence of Cas13b protein.

## Results

**Cas13b-independent effects with in vitro-transcribed guides.** We first aimed to determine if Cas13b was functional in different mosquito cells lines using transfected guide RNAs. Two guide RNAs were designed to target RNA sequences with different predicted RNA structures within the CHIKV nsP2 coding region (Supplementary Fig. 1, Supplementary Tables 1 and 2). The guides were generated by in vitro transcription. A plasmid expressing a reporter RNA containing the CHIKV target sequence and firefly luciferase coding sequence (pCHIKVLuc, Supplementary Fig. 2) was co-transfected into *Ae. aegypti* derived Aag2, AF05 and AF319 cells, and *Ae. albopictus* derived C6/36 cells with the guide RNAs and a second plasmid expressing Cas13b (pCas13b, Supplementary Figs. 2 and 3, Supplementary Table 3). AF05 and AF319 are cloned derivatives of Aag2 cells, with AF319 having a gene-edited knockout of *Dicer-2* (Dcr2)[21]; C6/36 is also deficient in Dcr2[22–24]. Each guide RNA (guide 1 and guide 2) was tested in two different quantities per transfection (10 and 40 ng, Supplementary Table 3).

In the presence of Cas13b, both guides (1 and 2) at both 10 and 40 ng reduced the expression of the CHIKV-luciferase reporter, relative to a *Renilla* luciferase control, in all *Ae. aegypti* cell lines (Fig. 2a–c, e–g, Supplementary Tables 4–6). In *Ae. albopictus* C6/36 cells there was a similar effect, with the exception of guide 1 at 40 ng where no difference from the control was observed (Fig. 2d, h, Supplementary Table 7). In all cell types the reduction in expression was greater with guide 2 than guide 1 (Fig. 2, Supplementary Tables 4–7). Unexpectedly, the guide-dependent knockdown was independent of the presence of the plasmid encoding Cas13b, as similar knockdown was observed when a plasmid expressing ZsGreen (pZsG, Supplementary Fig. 2) was used in place of the Cas13b plasmid (Fig. 2, Supplementary Tables 4–7).

**Partial Cas13b-dependent effects with U6-driven guides.** We speculated that the Cas13b-independent knockdown of gene expression might be a consequence of the use of in vitro-transcribed guide RNAs transfected into the cells. To test whether the source of the RNA affected the requirement for Cas13b we expressed the same guide RNAs under the control of an *Ae.*

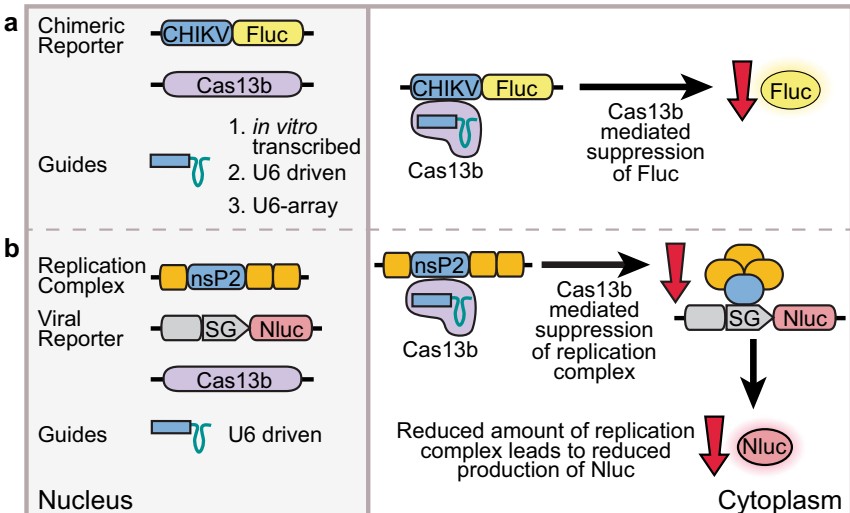

**Fig. 1 Strategy for assessment of Cas13b mediated suppression in insect cell lines.** In vitro-transcribed or U6-driven guides were used to assess if Cas13b is able to target viral sequences in **a** a directly targeted chimeric reporter, or **b** a viral reporter responding to targeted viral replicase. CHIKV: chikungunya virus sequence, Fluc: firefly luciferase, U6: *Ae. aegypti* U6-3 promoter, SG: CHIKV subgenomic promoter, Nluc: nanoluciferase.

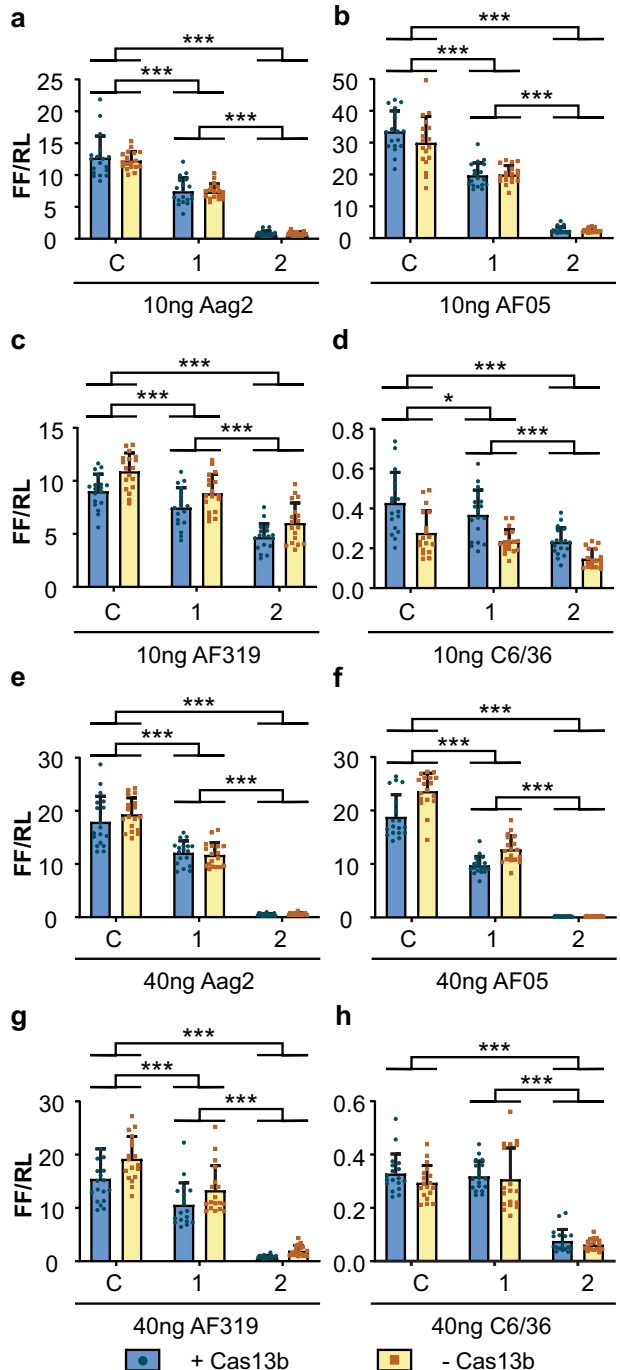

**Fig. 2 Cas13b-independent interference of a CHIKV-luciferase reporter in insect cells.** Aag2 (**a, e**), AF05 (**b, f**), Dcr2 knockout AF319 (**c, g**) and C6/36 (**d, h**) cells were co-transfected with plasmid expressing Cas13b (+ Cas13b) or ZsGreen (− Cas13b) and in vitro-transcribed guides at 10 ng (**a–d**) or 40 ng (**e–h**) per well. Each bar represents the mean values of firefly luciferase activity normalised to the *Renilla* luciferase activity (FF/RL), the whiskers are the standard deviation from three independent transfections with six replicates each and each dot represents a replicate. Linear mixed model, except for **h** two-way ANOVA, as model could not be fitted. ***$P < 0.001$, *$P < 0.05$ (Supplementary Tables 4–7). C: non-targeting control guide (AmC3), 1: guide 1, 2: guide 2.

*aegypti* U6 RNA promoter (Supplementary Fig. 4, Supplementary Table 3). In Aag2 cells, U6 promoter-driven guides were able to direct Cas13b-dependent knockdown of target gene expression. A similar amount of Cas13b-dependent knockdown was observed

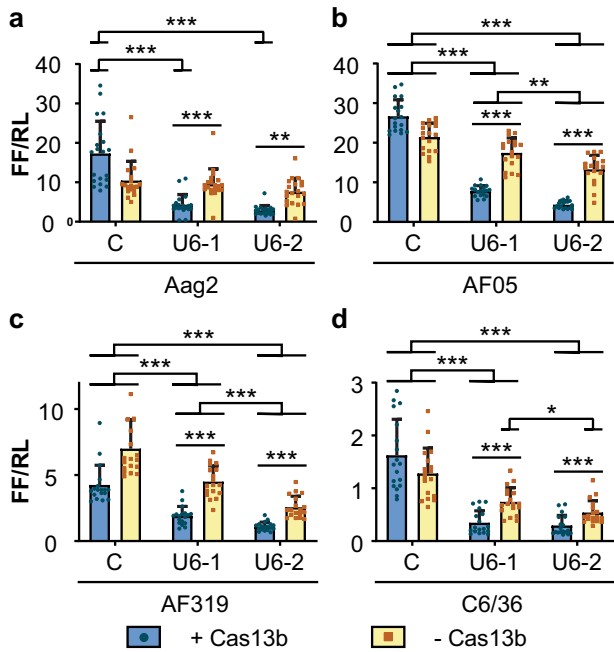

**Fig. 3 Partially Cas13b mediated interference of a CHIKV-luciferase reporter in insect cells.** Aag2 (**a**), AF05 (**b**), Dcr2 knockout AF319 (**c**) and C6/36 (**d**) cells were co-transfected with plasmid expressing Cas13b (+ Cas13b) or ZsGreen (− Cas13b) with U6 promoter-driven guide RNAs. Each bar represents the mean values of firefly luciferase activity normalised to the *Renilla* luciferase activity (FF/RL); the whiskers are the standard deviation from three independent transfections with six to eight replicates each and each dot represents a replicate. Linear mixed-effect model. ***$P < 0.001$, **$P < 0.01$ (Supplementary Tables 8–11). C: non-targeting control guide (U6-AmC3), U6-1: guide 1, U6-2: guide 2.

from both guides (Fig. 3a, Supplementary Table 8). In other mosquito cell lines knockdown of gene expression was only partially Cas13b-dependent, with knockdown enhanced by the presence of Cas13b (Fig. 3b–d, Supplementary Tables 9–11). In these cell lines Cas13b-independent knockdown of reporter gene expression was particularly noticeable with guide 2 and in AF319 and C6/36 cells (Fig. 3c–d, Supplementary Tables 10–11). Based on the mean ratios of each group, a reporter expression of 36.7 (SD ± 10.9)% and 47.7 (SD ± 16.2)% was observed with guide 2 in Cas13b-negative AF319 and C6/36 cells, respectively, in comparison to cells transfected with non-targeting control guide RNA (AmC3, Supplementary Table 2).

**Suppression of a CHIKV split replication system in vitro.** Having established that Cas13b could downregulate RNA with a reporter system we were interested to test whether it was able to affect replication of virus RNAs. Since direct work with CHIKV in the UK requires high containment facilities, we used a split replication system[19,20] in which the replicase polyprotein required for the replication of the viral RNA was supplied in *trans*. A plasmid expressing a modified virus genome in which the structural proteins were replaced by a nanoluciferase (Nluc) reporter (pCHIKVRep1, Supplementary Fig. 2) was co-transfected with plasmids expressing the viral replicase proteins (pCHIKVRep2, Supplementary Fig. 2), either Cas13b (pCas13b) or ZsGreen (pZsG), a firefly luciferase reference plasmid (pHr5Fluc, Supplementary Fig. 2), and a plasmid expressing U6 promoter-driven guide RNAs (Supplementary Fig. 5, Supplementary Table 3).

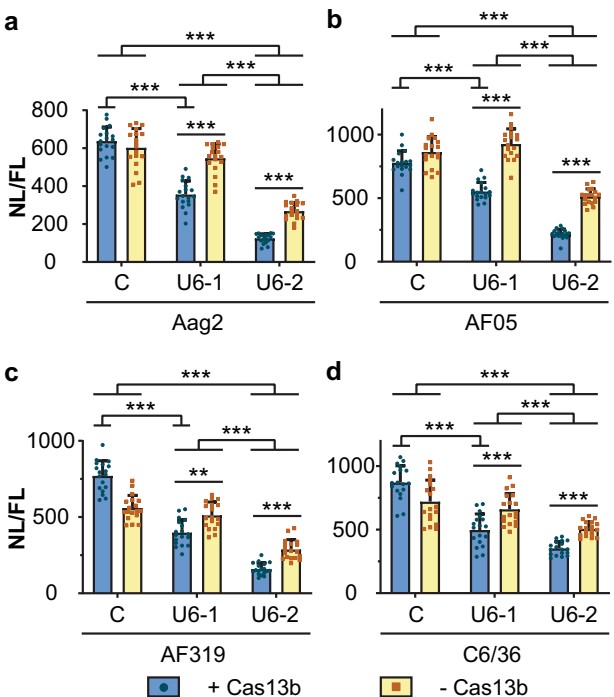

**Fig. 4 Effect of guide RNAs and Cas13b on the expression of a CHIKV split replication system. a** Aag2, **b** AF05, **c** Dcr2 knockout AF319 and **d** C6/36 cells were co-transfected with plasmid expressing Cas13b (+ Cas13b) or ZsGreen (− Cas13b) and U6 promoter-driven RNA guides, together with a split replication system with nanoluciferase under the control of a CHIKV subgenomic promoter, and a plasmid expressing firefly luciferase as control for transfection efficiency. Each bar represents the mean values of nanoluciferase activity normalised to the firefly luciferase activity (NL/FL), the whiskers are the standard deviation from three independent transfections with six replicates each and each dot represents a replicate. **a**, **b**, **d** Linear mixed-effect model, **c** two-way ANOVA as model could not be fitted. ***$P < 0.001$, **$P < 0.01$ (Supplementary Tables 12–15). C: non-targeting control guide (U6-AmC3), U6-1: guide 1, U6-2: guide 2.

Consistent with the non-replicating reporter experiments with U6 promoter-driven guides (Fig. 3), the presence of Cas13b and either guide significantly reduced expression of Nluc from the viral replication system in all cell types (Fig. 4, Supplementary Tables 12–15). In the case of guide 1, knockdown was accountable entirely by Cas13b-dependent viral RNA degradation. The Cas13b-transfected cells showed a significant drop in Nluc expression ($P < 0.001$, Fig. 4, Supplementary Tables 12–15) compared to the non-specific guide RNA control, but the controls without Cas13b (ZsGreen) did not ($P > 0.05$, Fig. 4, Supplementary Tables 12–15).

Also consistent with previous experiments, guide 2 resulted in a greater knockdown of viral replication system RNA levels. Guide 2 gave a significant reduction of Nluc expression alone ($P < 0.001$, Fig. 4, Supplementary Tables 12–15) and this effect was enhanced by the addition of Cas13b ($P < 0.001$, Fig. 4, Supplementary Tables 12–15) in all cell types.

**Suppression of a CHIKV reporter with multi-guide arrays**. A limitation of using sequence-specific knockdown of viral RNA is the potential for virus escape by small sequence changes in the guide RNA target site. This is particularly problematic for systems using relatively short targeting RNAs, such as shRNA, miRNA and CRISPR/Cas-based systems. A potential solution to this problem would be to express multiple guide RNAs inside the same cell, thereby simultaneously targeting multiple sequences

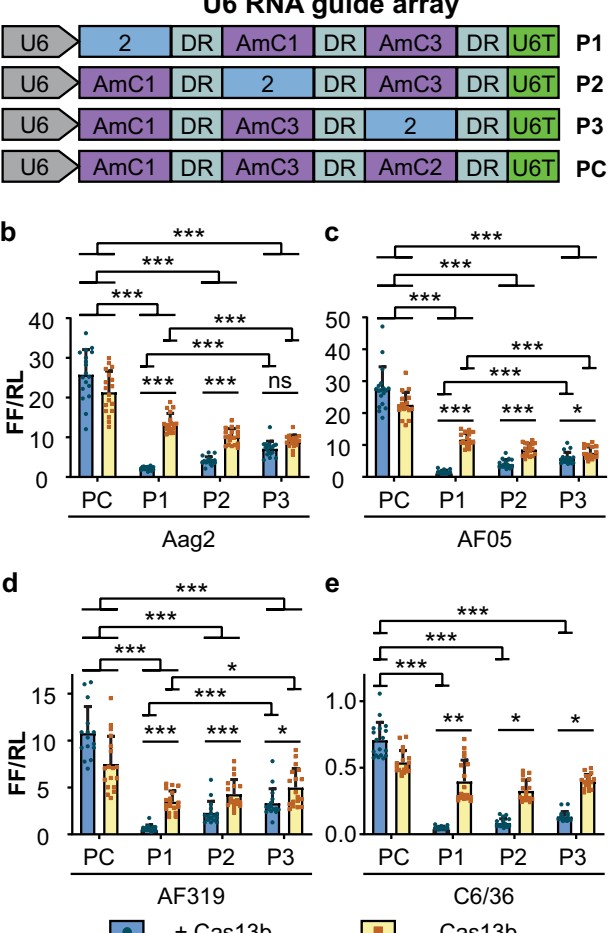

**Fig. 5 Use of a multi-guide array. a** Pol III promoter (U6)-driven RNA guide arrays used to test the effect of guide position on knockdown. Constructs P1, P2 and P3 have the active guide (guide 2) in positions 1, 2 and 3, respectively, of the three-position array, with guides against an absent target (AmCyan) in the remaining positions. A mock array (PC) with all the positions filled with non-targeting guides (AmC1, AmC2, AmC3) was used as control. **b–e** Interference of the CHIKV-luciferase reporter with U6 promoter-driven RNA array in Aag2 (**b**), AF05 (**c**), Dcr2 knockout AF319 (**d**) and C6/36 (**e**) cells. Each bar represents the mean values of firefly luciferase activity normalised to the *Renilla* luciferase activity (FF/RL); the whiskers are the standard deviation from three independent transfections with four to six replicates each and each dot represents a replicate. **b–d** Linear mixed effects model, **e** two-way ANOVA. ***$P < 0.001$, **$P < 0.01$, *$P < 0.05$, ns: not significant (Supplementary Tables 16–19). DR: non-variable RNA guide backbone, U6T: U6 promoter terminator, P1: position 1, P2: position 2, P3: position 3.

within the target RNA. In its native prokaryotic context, Cas13b processes longer RNAs to produce guides and has been shown to be capable of doing this in mammalian cells[12,13]; we therefore investigated whether our guide RNAs would function in the context of an array of guides.

We designed U6 promoter-driven guide arrays with three guide positions (Fig. 5a). One position in each array contained a CHIKV targeting guide (guide 2) while the other two positions had guides targeting sequences absent from the reporter (AmC1, AmC3, Fig. 5a, Supplementary Table 2). Plasmids expressing each of the arrays were co-transfected into mosquito cells with a plasmid expressing either Cas13b (pCas13b) or ZsGreen (pZsG), the CHIKV-luciferase reporter plasmid (pCHIKVLuc), and a

reference plasmid expressing *Renilla* luciferase (Supplementary Fig. 2, Supplementary Table 3).

Cas13b-dependent enhancement of suppression of the CHIKV-luciferase reporter was detected in all cell types (Fig. 5b–e, Supplementary Tables 16–19). Aag2 cells transfected with Cas13b showed greater reduction of luciferase expression compared to control cells transfected with the plasmid expressing ZsGreen where guide 2 was in position 1 or position 2 in the array, but no difference when it was in position 3 (Fig. 5b, Supplementary Table 16). In all other cell lines (AF05, AF319, C6/36) guide 2 was effective at mediating Cas13b-dependent knockdown of the CHIKV-luciferase reporter in all three array positions (Fig. 5c–e, Supplementary Tables 17–19). In Aag2, AF05 and AF319 cells, the knockdown effect was higher in position 1 compared to position 3 (Fig. 5b–d, Supplementary Tables 16–18) but in C6/36 there was no significant difference in the size of the reduction and the position of guide 2 in the array (Fig. 5e, Supplementary Table 19).

Consistent with the U6-driven single guide experiments with guide 2 (Fig. 3), we also observed Cas13b-independent knockdown in Aag2, AF05 and AF319 cells (Fig. 5b–d, Supplementary Tables 16–18). This effect was greater when the specific guide was in position 1 compared to position 3 in Aag2 and AF05 cells (Fig. 5b–c, Supplementary Tables 16–17), while the opposite (lower in position 1 compared to position 3) was detected in AF319 cells (Fig. 5d, Supplementary Table 18). No Cas13b-independent interference was observed in C6/36 cells (Fig. 5e, Supplementary Table 19).

## Discussion

In this study we have shown that Cas13b RNA guides are able to induce the knockdown of target genes in mosquito cells, both through Cas13b-mediated enhancement of suppression and, unexpectedly, through Cas13b-independent, sequence-specific, RNA knockdown. This is in contrast to work describing the use of Cas13b guides in mammalian and plant cells where RNA knockdown was entirely dependent on the presence of Cas13b[12,17]. We hypothesise that the phenomenon is dependent on base-pair complementarity between the guide and the target RNA, leading to recognition and processing of the target RNA by cellular RNA interference (RNAi) systems. To further investigate this, we used two Dcr2-defective cell lines, C6/36 and AF319. Guide-only suppression was observed in these cell lines to a similar degree to that observed in Aag2 and AF05 cells. We, therefore, conclude that the guide-only suppression is not substantially dependent on Dcr2.

We used the chimeric CHIKV-luciferase reporter to assess if Cas13b was functional in mosquito cells (Fig. 1). In these experiments, expression of the CHIKV-luciferase reporter was significantly reduced in the presence of Cas13b plus anti-CHIKV guide RNA, or guide alone, relative to a *Renilla* luciferase control (Fig. 3, Supplementary Fig. 4). This knockdown of the firefly luciferase in the dual-luciferase assay indicates a reduction in the amount of firefly luciferase mRNA present, as showed elsewhere for RNAi[25], consistent with direct targeting of the CHIKV-luciferase mRNA by Cas13b or guide RNA.

The Cas13b-independent effect was detected due to the use of a ZsGreen mock control plasmid (pZsG). Previous work in plant and mammalian cells has not reported interference by the guides alone[12,17]. In plants, the targeting guides were used without Cas13b[17], while in mammals, the authors used a catalytically inactive dCas13b[12]. We speculate that the difference with our results may be due to the cell type used, and that the dCas13b may bind and sequester the guides such that these are less available to endogenous RNAi pathways. The ability of dCas13

enzymes to bind to guides and recognise targets has been demonstrated in the context of programmable RNA editing and tracking of RNA in living cells[26–28].

The degree of Cas13b-independent suppression varied between different guides and delivery methods. For example, with transfection of in vitro-transcribed guides suppression appeared entirely Cas13b-independent (Fig. 2), whereas with U6 guides addition of Cas13b generally enhanced suppression; in some experiments, guide 1 alone did not show significant suppression in the absence of Cas13b in comparison to guide 2 (Fig. 4).

The differences observed between the guides may be due to several reasons. Based on predictions from RNAup[29], which can predict RNA accessibility based on RNA structural opening energies, target site 2 is more accessible than target site 1 (Supplementary Table 1). Cas13 activity is also dependent on the base composition at the surrounding target site[30]; Cas13b from *Prevotella sp.* is reported to prefer open A-rich motifs for cleavage[18,31].

Cas13b-independent suppression is likely occurring in each case, but not always to an extent detectable in our experiments with the weaker guide 1. It is more surprising that addition of Cas13b did not increase suppression with in vitro-transcribed guides; possibly the Cas13b-independent effect was saturating, though suppression with guide 1 was not particularly strong in these experiments (Fig. 2). This emphasises the importance of designing multiple guides, perhaps informed by bioinformatics analyses such as RNA structure accessibility and testing the efficacy of each guide in the presence and absence of Cas13b.

A possible explanation for the Cas13b-independent RNA knockdown effect observed in our experiment is that double-stranded RNA regions in the guide RNAs were recognised by the cellular RNAi system. Since insects are heavily reliant on this system as the primary adaptive response against viruses with RNA genomes[32,33], it is possible that insect cells would be more sensitive to small highly structured RNA triggers than mammalian cells. This is consistent with the observation that introducing in vitro-transcribed guides into the cytoplasm was more potent at stimulating Cas13b-independent knockdown than transcribing the guides from an RNA polymerase III promoter in the nucleus (Figs. 2 and 3), though the level of guides produced in those experiments may not be equivalent.

The sequence-specific silencing observed in cells not expressing Cas13b was not expected and the mechanism is not clear. It is possible that Dcr2 recognises the stem–loop structures of the guide RNA backbone and directs these to the Dcr2 small interfering RNA pathway. However, as Dcr2 knockout AF319 cells also showed some limited Cas13b-independent silencing, other alternatives should be considered, such as a direct loading of single-stranded RNA guides onto Ago1 and Ago2, the catalytic components of RNA-induced silencing complexes (RISC)[34–37]. It may be that the short, highly structured RNAs that are recognised by Cas13b are loaded efficiently onto the RISC complexes even in the absence of Dcr2 in insect cells. Indeed, recombinant mammalian Ago2 has been reported to bind to single-stranded RNAs up to 73 nt long[34]. Another explanation could be possible redundancies between Dcr2-mediated small interfering RNA, Dcr1-mediated microRNA and the less known Piwi-interacting RNA pathways, where interactions between these pathways have been described[21,38–41] and components of the latter two pathways have been shown to be involved in antiviral responses as well[21,42–45].

Although U6-driven guides were previously used in human and plant cells[26], we were uncertain whether expressing Cas13b guides from a U6 promoter would be effective in insect cells. Normally, U6 RNAs are retained in the nucleus, whereas Cas13b is translated in the cytoplasm and presumably acts there against

viral RNAs. Despite this theoretical concern, our data with the U6-driven guides demonstrate that Cas13b-independent and Cas13b-dependent suppression could be achieved with U6-driven guides (Figs. 3–5). These data also indicate that the γ-monomethyl cap typical of pol III-transcribed U6 RNAs[46] does not prevent these activities.

Given standing genetic variation and mutation, a single guide RNA is unlikely to provide adequate knockdown for all members of a large population. This applies to mosquito genes but even more so to RNA viruses, which have high mutation rates and the potential for rapid evolution of escape mutants. Use of guide RNAs for engineered refractoriness against such viruses is therefore likely to require the use of multiple guides. One of the attractive features of Cas13b is that it can process its guides from a longer RNA[13], which we demonstrated, using a chimeric CHIKV-luciferase reporter (Fig. 5). We also assessed the effect of positions in an array. We found strong suppression using a targeting guide in either the first, second or third position in a three-guide array. However, suppression was somewhat reduced as we moved the guide more 3′. Since the most 5′ position will produce a capped guide and the others will not, this is further evidence that the cap is at least not detrimental to suppression. One possible explanation for the modest reduction in effectiveness in more 3′ positions is that the sequence encoding the guide RNA backbone includes four consecutive T nucleotides (Supplementary Table 1). Pol III transcription of U6 sequences terminates at homopolymeric runs of T; $T_4$ has been shown to provide partial termination in mammalian cells[47]. It therefore seems plausible that the reduced effectiveness from more 3′ positions simply reflects reduced synthesis of the guide in these positions. It may be that this sequence could be mutated to overcome this, similar to work with Cas9 guide RNAs[48], or a different Cas13 or guide combination used. Another possibility is that Cas13b is thought to process its guide array from 3′ to 5′[13,49,50], which may affect the relative utilisation of each guide.

A CHIKV split replication system[19,20] was used to assess if suppression of viral replication could be achieved (Fig. 1). In this system, viral replicase leads to expression of a reporter from a subgenomic promoter on a virus-derived RNA from which the replicase coding sequences have been deleted. Reduction in viral replicase, e.g. through targeting nsP2 mRNA coding sequence, leads to reduction in reporter activity (Fig. 4, Supplementary Fig. 5). A somewhat lesser degree of target interference was observed with the split replication system (Fig. 4) in comparison to the chimeric CHIKV-luciferase (Fig. 3), possibly due to fully functional replication complexes produced before Cas13b interference, which would lead to basal amplification of the reporter.

The split replication system is a model for CHIKV; we consider that it reproduces the relevant elements with respect to Cas13b but it is possible that subtle differences mean that the effect, or the magnitude of the effect on CHIKV itself might be rather different. For example, the replicase RNAs are generated in the nucleus by the Pol II polymerase, whereas viral RNAs are generated by virus-encoded replicase in the cytoplasm; a nucleus-specific effect might therefore show differential activity. On the other hand, mutations identified with this split replication system have been shown to have similar effects when introduced into replicating virus[19]. Also, Cas13b-mediated RNA virus inhibition has been demonstrated in both human and plant cells[17,18].

We show here that Cas13b can be used to suppress a CHIKV split replication system[19,20]. This and the viral inhibition observed in other species[17,18] suggests that Cas13b provides a potential approach to engineering virus refractoriness in mosquitoes. Other Cas13 variants are also known, with broadly comparable properties but potentially more efficient for specific applications[14,17]. That guide RNAs have suppressing effects on

their own is interesting but unlikely to reduce their utility for engineering refractoriness. Cas13 provides an alternative method to targeting virus sequences rather than RNAi-based methods, which have previously been shown to be potentially effective against CHIKV, Zika and dengue viruses[51–54]. Sequence-based approaches have advantages of specificity, but corresponding difficulties in targeting multiple viruses, or even all sequence variants of a given virus, and correspondingly may be prone to resistance through the emergence of sequence variants. Similar issues of resistance evolution likely apply to the use of monoclonal antibodies as the basis for virus resistance[55]. One way to overcome such issues is by multiplexing—simultaneously targeting multiple sequences in the same or different viruses. Cas13b has some advantages in the relative ease of multiplexing, particularly in its ability to process guide RNAs from a compact array on a single primary transcript. We tested this potential by placing an active guide at each of three positions in a three-element array and found it to be effective in all positions (Fig. 5), though with a modest reduction in effectiveness in more 3′ positions that may relate to premature termination of transcription. Taken together, these data are very encouraging for the future development of antiviral effectors in mosquitoes based on RNA-directed Cas systems.

## Methods

**Plasmid construction.** AGG1186:pHr5Fluc (Genbank accession no. MT119956, Supplementary Fig. 2), expressing firefly luciferase (Fluc) under the control of the baculovirus *Hr5-ie1* promoter, was made by modifying the pGL3 vector (Promega) to insert a *Hr5-ie1* promoter[56], an intron from the *Drosophila melanogaster alcohol dehydrogenase* (*adh*) gene[57] and the Kozak consensus sequence[58] upstream of the coding region of Fluc. In all, 200 bp of the nsP2 sequence of CHIKV LR2006-OPY1 strain (ECSA genotype) were added to the 5′ end of Fluc with a ubiquitin fusion[59] to build AGG1221:pCHIKVLuc (Genbank accession no. MT119958; Supplementary Fig. 2), which expresses the CHIKV-luciferase reporter.

To construct the Cas13b-expressing plasmid (AGG1328:pCas13b, Genbank accession no. MT119959; Supplementary Fig. 2), the PspCas13b sequence of *Prevotella sp. P5-125*[12] was codon optimised for *Ae. aegypti*, synthesised in two fragments (Twist Bioscience) and assembled with assembly PCR using Q5 high-fidelity DNA polymerase (New England BioLabs). Cas13b was cloned into a vector containing an ampicillin-resistance gene such that Cas13b was under the control of the constitutive *Hr5-ie1* promoter. A T2A ribosomal stutter was inserted on the 3′ end of Cas13b to attach ZsYellow marker protein for visualisation.

A plasmid expressing ZsGreen fluorophore under the control of the *Hr5-ie1* promoter (AGG1201:pZsG, Genbank accession no. MT119955, Supplementary Fig. 2) was synthesised as a mock control for pCas13b.

The AGG1080:pRL-Opie2 *Renilla* luciferase plasmid (Supplementary Fig. 2) was provided by M. Anderson[60] and was used as a reference plasmid to normalise data for transfection efficiency.

*Aedes aegypti* U6-3 Pol III promoter was amplified from *AeU6-702* (AGG1120)[60] with primers 2704 and 2705 (Sigma-Aldrich; Supplementary Table 20). The non-variable stem–loop backbone for the CRISPR-RNA of PspCas13b (Cas13b)[12] with *BsaI* restriction sites on each end to allow the insertion of different RNA guide sequences was synthesised by assembly PCR using primers 2706 and 2707 (Supplementary Table 20). These two PCR products were inserted into the pJet1.2 vector (Thermo Scientific) by HiFi assembly (New England BioLabs) to build plasmid AGG1399, the U6-guide vector (Genbank accession no. MT119950; Supplementary Fig. 2). Oligos for the variable RNA guide sequences (primers 2712 and 2713 for guide 1, and primers 2714 and 2715 for guide 2; Supplementary Table 20) were annealed and inserted into BsaI-linearised AGG1399 to build the different U6-single guide plasmids (AGG1879:U6-1, Genbank accession no. MT119960, and AGG1880:U6-2, Genbank accession no. MT119961). An AmCyan-targeting guide (AmC3; Supplementary Table 2) was designed as the negative control and cloned into the same vector (primers 2720 and 2721, AGG1881:U6-AmC3, Genbank accession no. MT119962; Supplementary Table 20).

For the U6-driven guide array, two additional guides targeting AmCyan were designed (AmC1 and AmC2; Supplementary Table 2). AmC1 and AmC3 were used for positions that were not occupied by the CHIKV targeting guide 2 (Fig. 5a). A mock array with guides AmC1, AmC2 and AmC3 (Fig. 5a) was built as a control (AGG1875:U6-PC; Genbank accession no. MT119951). The arrays were assembled with three variable RNA guide sequences, each followed by the non-variable backbone, into the AGG1399 vector, such that the expression of the array was controlled by the U6-3 promoter. Guide 2 was inserted into positions one, two or three in AGG1876:U6-P1 (Genbank accession no. MT119954), AGG1877:U6-P2

(Genbank accession no. MT119953) and AGG1878:U6-P3 (Genbank accession no. MT119952), respectively, to test the activity of each position.

For the CHIKV split replication system, the plasmids Ubi-P1234-CHIKV (pCHIKVRep2; Supplementary Fig. 2) and Ubi-Fluc-Gluc plasmids have been previously described[19,20]. AGG1521:pCHIKVRep1 (Genbank accession no. MT119957; Supplementary Fig. 2) was derived from Ubi-Fluc-Gluc by replacing Fluc and Gluc with the fluorophore EGFP and Nluc, respectively.

**In vitro-transcribed RNA guides and arrays**. The nsP2 region of CHIKV was targeted; this encodes the viral protease responsible for processing the viral polyprotein into its functional components. Target structures and accessibility were predicted with Mfold[61] and RNAup[30]. Nucleotide constraints, protospacer flanking site motifs, for target selection have not been reported for PspCas13b[12]. No homologous targets in the RNA transcripts of *Ae. aegypti* (AaegL3 and AaegL5) and *Ae. albopictus* (AaloF1) were found[62,63] for each of the RNA guides designed.

DNA templates of single RNA guides for PspCas13b (guide 1, guide 2 and AmC3 control) were synthesised by PCR using Phusion high-fidelity DNA polymerase (New England BioLabs), a guide-specific 71–72 nt forward primer (10 nM; Sigma-Aldrich) containing a T7 promoter, variable guide sequence and the first 18 nt of the non-variable stem–loop guide backbone as previously published[12] (primers 2430, 2433 and 2946 for guide 1, guide 2 and AmC3 control, respectively; Supplementary Table 20), and a 36 nt reverse complement primer (10 nM, Sigma-Aldrich) coding for the guide backbone (primer 2323, Supplementary Table 20). In vitro transcription and subsequent purification of guide RNAs was performed with the MEGAscript T7 transcription kit (Invitrogen) and the MEGAclear transcription clean-up kit (Invitrogen), respectively.

**Cell culture**. Aag2 cells (*Ae. aegypti* derived, RRID:CVCL_Z617) and C6/36 cells (*Ae. albopictus* derived, RRID:CVCL_Z230) which have a truncated Dcr2 that is not functional[22,24,64] were used for transfections. *Ae. aegypti* Dcr2 CRISPR knockout AF319 cells and its single-cell-derived parental cell line AF05 were obtained from K. Maringer, University of Surrey, United Kingdom[21,44,65]. All cell lines were authenticated by COI amplification and sequencing.

All cell lines were grown in Leibovitz-15 media (Gibco) supplemented with 10% foetal bovine serum (Labtech), 100 IU/ml penicillin with 100 μg/ml streptomycin (Gibco) and 10% tryptose phosphate broth (Gibco) at 28 °C without $CO_2$ or humidity control.

**Cell transfections**. Cells were seeded into 96-well microplates (Nunc) at a cell density of $5 \times 10^5$ cells/well. When cells reached 70–80% confluence, they were transfected using TransIT PRO reagent (Mirus Bio) according to the manufacturer's instructions. For each experiment, three independent transfections with 6–8 replicates were performed. pRL-Opie2 or pHr5FLuc (Supplementary Fig. 2) was used as transfection control where applicable. Amounts of plasmid and RNA mixtures used for transfections were optimised such that the total mass did not exceed 250 ng. A guide targeting AmCyan, a gene not present in the reporter RNA, was used as a control to test the specificity of the effect and a plasmid expressing the fluorescent reporter ZsGreen (pZsG; Supplementary Fig. 2) was used in place of the plasmid-expressing Cas13b (pCas13b, Supplementary Fig. 2) to determine if any knockdown observed was Cas13b dependent without changing the total molar mass of DNA transfected. Specific details for transfection mixtures are listed in Supplementary Table 3. Media were replaced with unsupplemented media before transfection. Cells were incubated for 3–5 h post-transfection at 28 °C before media were replaced with supplemented media. Cells were harvested for dual-luciferase assay after 2 days.

**Luciferase assays**. Luciferase assays were performed as previously described[60] with the Dual-Luciferase Reporter Assay (Promega) on a GloMax multi+ plate reader (Promega). Briefly, cells were lysed with 30 μl 1× passive lysis buffer (Promega) each after two washes with ion-free phosphate buffered saline (1× PBS) and subjected to one freeze–thaw cycle at −80 °C. Luciferase assay reagent II (Promega) and stop & glo reagent (Promega) were prepared according to the manufacturer's instructions and the luciferase activities were measured.

**Statistics and reproducibility**. In the experiments using pCHIKVLuc (Supplementary Fig. 2, Supplementary Table 3), firefly luciferase activity was normalised against *Renilla* luciferase activity to correct for transfection efficiency in each well (FF/RL). In the experiments with the CHIKV split replication system (Supplementary Fig. 2, Supplementary Table 3), the Nluc activity was normalised against firefly luciferase activity (NL/FF). The FF/RL or NL/FF ratios were analysed with R (version 3.6.1), and wells that were not transfected were not included. For each experiment, three independent infections with six replicates were performed. Linear mixed-effect models fit by restricted maximum likelihood were preferentially used to investigate the effect of Cas13b and its associated guides on the expression levels of the reporters used. The lmer function in the lme4 package[66] was used and the lmerTest package[67] was used to calculate the *t*-tests using Sattherthwaite approximations to degrees of freedom. Within each experiment, data were transformed as appropriate to fit a normal distribution and models were run separately for each cell line and guide concentration, where applicable. The presence of Cas13b and the guide used were set as categorical fixed factors and the experiment as a random factor. Stepwise deletion of

non-significant ($P > 0.05$) variables was used to build the final model where all variables were significant ($P < 0.05$). Diagnostic plots of residuals were checked to ensure that there was constant variance between residuals and model assumptions were met. Two-way ANOVA was used where a model could not be fitted. Tukey's honest significant differences test (multcomp package[68]) was used for post hoc analyses. Graphs were plotted and arranged with GraphPad Prism 8 and Illustrator CS6 (Adobe).

**Reporting summary**. Further information on research design is available in the Nature Research Reporting Summary linked to this article.

## Data availability
The data that support the findings of this study are available in Figshare at https://doi.org/10.6084/m9.figshare.c.4867740[69]. Sequence data of plasmids used in this study have been deposited in Genbank with the accession codes MT119950 to MT119962.

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

## Acknowledgements

The authors thank Michelle Anderson for pRL-Opie2 and AeU6-702 plasmids, Kevin Maringer for AF319 and AF05 cell lines, and Yu-mei Chang and Simon Gubbins for advice in the statistical analysis. P.Y.L.T., L.C.P., P.T.L. and L.A. were supported by Wellcome Trust Investigator Award 110117/Z/15/Z. S.A.N.V., J.P. and L.A. are supported by funding from the Biotechnology and Biological Sciences Research Council (BBSRC) [BB/M011224/1, BBS/E/I/00001985, BBS/E/I/00007033, BBS/E/I/00007038 and BBS/E/I/00007039] and A.M., R.F. and L.A. by Wellcome Trust/Newton Fund-MRC Collaborative Award 200171/Z/15/7.

## Author contributions

Conceived and designed the experiments: P.Y.L.T., L.C.P., R.F., A.M., P.T.L., R.N. and L.A. Performed the experiments and analysed the data: P.Y.L.T. Designed and generated constructs or components: P.Y.L.T., L.C.P., J.P. and S.A.N.V. Contributed reagents/materials: P.Y.L.T., L.C.P., S.A.N.V., A.M. and R.F. Wrote the paper: P.Y.L.T., L.C.P., R.N. and L.A. All authors read and approved the final draft.

## Competing Interests

The authors declare no competing interests.
