## [Peer Review File · Communications Biology]

Reviewers' comments:

Reviewer #1 (Remarks to the Author):

The author Priscilla Y. L. Tng et.al. introduced some about Cas13b system research in the manuscript "Cas13b-dependent and Cas13b-independent RNA knockdown of viral sequences in mosquito cells following guide RNA expression". The manuscript tries to demonstrate that Cas13b-dependent and Cas13b-independent is capable of mediating the specific degradation of viral RNA in mosquito cells. The manuscript, however, does not clearly show that the guide RNAs are able to induce degradation in the absence of Cas13b protein. Based on the present manuscript, therefore, we suggest review and resubmit manuscript before rewriting it.

1. Compared with elaborating the editing effect of cas13b, it should focus more on the analysis of the mechanism of RNA target without cas13b? This is more biologically meaningful.
2. Is this mechanism of RNA degradation without cas13b universal? In other animal cells or human cells? As you know, the Cas13b system has been proposed to target viral RNA in plants and mammals.
3. This degradation system that does not depend on cas13, besides RNAi, is there any other degradation system that depends on it? Another protease involved?
4. In all cell types the reduction in expression was greater with guide 2 than guide 1, what is the relationship with the target site design should be clearly stated.
5. Does this system, like the transcription mechanism of RNA virus in cells, have the ability to detect the virulence of the virus and analyze its inhibition?
6. The author should add direct evidence to prove that the RNA level is degraded, such as RT-PCR.

Reviewer #2 (Remarks to the Author):

The work by Tng et al. entitled "Cas13b-dependent and Cas13b-independent RNA knockdown of viral sequences in mosquito cells following guide RNA expression" describes the application of the CRISPR-Cas13 machinery to mediate targeted gene knockdowns. Surprisingly, the sgRNAs alone resulted in knockdown as well of the target genes.

The work is timely and the conclusions are important for the readers in CRISPR field and the RNA interference applications in general. The manuscript is well written and presented to the readers in the field.

I have one concern on this work regarding the use of virus sequences and not actually the virus itself. When the virus sequences are produced by the Pol II they cannot mimic the actual virus infection. And there might be certain mechanism at play enabling CRISPR systems to distinguish viral or foreign sequences from the self-transcripts generated by the host transcription machinery. Although the split system is slightly better but again it does not mimic an actual virus infection which is necessary to assess the validity of this approach for virus interference.

By the same token, efficiencies of CRISPR-Cas13 systems would vary between targeted knockdown of the host transcripts or the virus or pathogen transcripts. This point needs to be sufficiently discussed and elaborated in the manuscript.

There could be other mechanisms inhibiting the CRISPR-Cas13b activities, and also other CRISPR-systems including Cas13d or CasRx could be more efficient in targeted virus interference. And please cite recent work by Mahas et al. (Genome Biology) comparing 9 variants of Cas13 proteins for their activities in planta and showing that Cas13d exhibited robust virus interference.

In conclusion:

I understand that it may be complicated to work with these viruses. However, the authors need to sufficiently discuss these and other possibilities in the manuscript. And they are requested to have an

introductory sketch summarizing the experiments, molecular details of virus interference, and expected results when it does or does not work. This will make it easy for the readers to grasp the concept. Please do cite important work in the field including the ones I referred to here. And sufficiently discuss what other mechanisms at play could be mediating the knockdown via cas13-independent mechanisms

Point-by-point response to reviewers for the manuscript “Cas13b-dependent and Cas13b-independent RNA knockdown of viral sequences in mosquito cells following guide RNA expression” (COMMSBIO-20-0646A)

We thank the reviewers for their valuable comments for us to improve the manuscript. We have now incorporated these into the revised manuscript and the changes are listed in the table below.

Reviewer #1	Reply
The author Priscilla Y. L. Tng et.al. introduced some about Cas13b system research in the manuscript “Cas13b-dependent and Cas13b-independent RNA knockdown of viral sequences in mosquito cells following guide RNA expression” . The manuscript tries to demonstrate that Cas13b-dependent and Cas13b-independent is capable of mediating the specific degradation of viral RNA in mosquito cells. The manuscript, however, does not clearly show that the guide RNAs are able to induce degradation in the absence of Cas13b protein. Based on the present manuscript, therefore, we suggest review and resubmit the manuscript before rewriting it.	We thank reviewer 1 for the comments and have now revised the manuscript accordingly. Details relevant to each comment are listed below.
1. Compared with elaborating the editing effect of cas13b, it should focus more on the analysis of the mechanism of RNA target without cas13b? This is more biologically meaningful.	We agree that it is interesting that the Cas13b guides seem to be able to stimulate Cas13b independent suppression of gene expression. To address this point, we have revised the discussion to include more information about possible mechanisms involved in RNA silencing without Cas13b (Lines 179-197).
2. Is this mechanism of RNA degradation without cas13b universal? In other animal cells or human cells? As you know, the Cas13b system has been proposed to target viral RNA in plants and mammals.	No Cas13b independent silencing has been reported in mammals or plants. We have now included this information (Ref. 13 and 18) in the discussion section (Lines 157-162). The controls used by the authors in these other studies were different to the one used in this paper, and the potential implication of these differences are also now included in the discussion in the revised manuscript.
3. This degradation system that does not depend on cas13, besides RNAi, is there any other degradation system that depends on it? Another protease involved?	To uncover the degradation system/systems involved is really interesting, but beyond the scope of the current work. We would note that the degradation is sequence specific because the reference reporter was unaffected. This would seem to rule out the idea that degradation is from a non-specific protease effect. We have revised the discussion section to include more information regarding possible Cas13 independent RNA degradation systems (Lines 188-197).
4. In all cell types the reduction in expression was greater with guide 2 than guide 1, what is	We have included an analysis of predicted accessibility of the target sites (Line 301,

the relationship with the target site design should be clearly stated.	Supplementary Table 1) and a discussion about the differences between the two guides (Lines 168-172) in the revised manuscript.
5. Does this system, like the transcription mechanism of RNA virus in cells, have the ability to detect the virulence of the virus and analyze its inhibition?	The split replication system generates viable viral replication complex that mimics the infection process and transcription of viral mRNA in a cell but lacks the ability to infect neighbouring cells and hence cannot be used to model the cell to cell spread of the virus. The limitations of the split replication system and potential implications for the conclusions of this manuscript are now discussed (Lines 223-238).
6. The author should add direct evidence to prove that the RNA level is degraded, such as RT-PCR.	We have now included a new reference and paragraph in the discussion section addressing the relation between the results obtained with a dual luciferase assay and RT-PCR (Lines 150-155).
Reviewer #2	Reply
The work by Tng et al. entitled “Cas13b-dependent and Cas13b-independent RNA knockdown of viral sequences in mosquito cells following guide RNA expression” describes the application of the crispr-cas13 machinery to mediate targeted gene knockdowns. Surprisingly, the sgRNAs alone resulted in knockdown as well of the target genes. The work is timely and the conclusions are important for the readers in CRISPR field and the RNA interference applications in general. The manuscript is well written and presented to the readers in the field.	Comments from reviewer 2 are much appreciated, and we have revised the manuscript accordingly.
1. I have one concern on this work regarding the use of virus sequences and not actually the virus itself. When the virus sequences are produced by the Pol II they cannot mimic the actual virus infection. And there might be certain mechanism at play enabling CRISPR systems to distinguish viral or foreign sequences from the self- transcripts generated by the host transcription machinery. Although the split system is slightly better but again it does not mimic an actual virus infection which is necessary to assess the validity of this approach for virus interference.	We thank the reviewer for the concerns and understand the limitations of the split replicon system to study virus interference. Certainly, the reporter systems we use are models of virus infection and have some differences from ‘the real thing’, though they are very closely related and we think reproduce all the key aspects. One difference is that Pol II transcripts are generated in the nucleus then exported to the cytoplasm for (e.g.) translation, whereas the virus produces its RNA in the cytoplasm. However, Cas13b lacks a visible nuclear localisation signal and we have no reason to think that key Cas13b processes take place in the nucleus. We have now addressed the limitations of the split replicon system as a mimic of virus RNA in the discussion (Lines 223-238).

2. By the same token, efficiencies of CRISPR-Cas13 systems would vary between targeted knockdown of the host transcripts or the virus or pathogen transcripts. This point needs to be sufficiently discussed and elaborated in the manuscript.	We have now addressed this point in the revised discussion section (Lines 223-238).
3. There could be other mechanisms inhibiting the CRISPR-Cas13b activities, and also other CRISPR- systems including Cas13d or CasRx could be more efficient in targeted virus interference. And please cite recent work by Mahas et al. (Genome Biology) comparing 9 variants of Cas13 proteins for their activities in planta and showing that Cas13d exhibited robust virus interference.	We thank the reviewer for their comments. The existence of other Cas13 variants is now addressed in lines 40-41 and 241-242, and the important work of Mahas et al. is cited and discussed (Lines 43, 143-145, 157-159, 237-238, 241-242).
4. In conclusion: I understand that it may be complicated to work with these viruses. However, the authors need to sufficiently discuss these and other possibilities in the manuscript.	We thank the reviewer for understanding and have now discussed this topic in lines 223-233.
5. And they are requested to have an introductory sketch summarizing the experiments, molecular details of virus interference, and expected results when it does or does not work. This will make it easy for the readers to grasp the concept.	This suggestion is much appreciated. We have now included a new introductory figure 1 to the manuscript (Lines 513-518). This figure and its legend are also included at the bottom of this document.
6. Please do cite important work in the field including the ones I referred to here.	We have now included the references highlighted by the reviewer (References 15, 18, 29, 31).
7. And sufficiently discuss what other mechanisms at play could be mediating the knockdown via cas13-independent mechanisms	We have added to the discussion section what other mechanisms could be involved in the Cas13b independent silencing observed (Lines 179-197).

Figure 1: Strategy for Cas13b mediated suppression of a (a) chimeric firefly luciferase reporter (chimeric reporter) and a (b) CHIKV split replication system (replication complex and viral reporter). *In vitro* transcribed or U6 driven guides were used to assess if Cas13b is able to target viral sequences in insect cell lines. (a) The chimeric reporter was directly targeted, while (b) the viral reporter, recognised and produced by the viral replication complex, was suppressed by reducing the amount of replication complex. CHIKV: chikungunya virus sequence, Fluc: firefly luciferase, U6: *Ae. aegypti* U6-3 promoter, SG: CHIKV subgenomic promoter, Nluc: nanoluciferase.